Amino-acid site variability among natural and designed proteins

Jackson Eleisha L. 1
Ollikainen Noah 2
Covert Arthur W. III 1
Kortemme Tanja 2 3
Wilke Claus O. 1 wilke@austin.utexas.edu
1 Institute of Cellular and Molecular Biology, Center for Computational Biology and Bioinformatics, and Department of Integrative Biology, The University of Texas at Austin , Austin, TX , USA
2 Graduate Program in Bioinformatics, University of California San Francisco , San Francisco, CA , USA
3 California Institute for Quantitative Biosciences (QB3) and Department of Bioengineering and Therapeutic Sciences, University of California San Francisco , San Francisco, CA , USA
Uversky Vladimir
Electronic publication date: 2013 Nov 12
Publication date: 2013
Volume: 1
Electronic Location ID: e211
Received 2013 Oct 3; Accepted 2013 Oct 24
Copyright: © 2013 Jackson et al.
Copyright year: 2013
Copyright holder: Jackson et al.
License: This is an open access article distributed under the terms of the Creative Commons Attribution License, which permits unrestricted use, distribution, and reproduction in any medium, provided the original author and source are credited.
License URL: https://creativecommons.org/licenses/by/3.0/

Keywords: Protein design, Fixed-backbone design, Flexible-backbone design, Sequence alignments, Relative solvent accessibility, Site variability

Funding: NIH grant R01 GM088344 DTRA grant HDTRA1-12-C-0007 NSF grant CAREER MCB-0744541 NSF grant DBI-1262182 This work was supported in part by NIH grant R01 GM088344 and by DTRA grant HDTRA1-12-C-0007 to COW and by NSF grants CAREER MCB-0744541 and DBI-1262182 to TK. The funders had no role in study design, data collection and analysis, decision to publish, or preparation of the manuscript.

==============================
Computational protein design attempts to create protein sequences that fold stably into pre-specified structures. Here we compare alignments of designed proteins to alignments of natural proteins and assess how closely designed sequences recapitulate patterns of sequence variation found in natural protein sequences. We design proteins using RosettaDesign, and we evaluate both fixed-backbone designs and variable-backbone designs with different amounts of backbone flexibility. We find that proteins designed with a fixed backbone tend to underestimate the amount of site variability observed in natural proteins while proteins designed with an intermediate amount of backbone flexibility result in more realistic site variability. Further, the correlation between solvent exposure and site variability in designed proteins is lower than that in natural proteins. This finding suggests that site variability is too uniform across different solvent exposure states (i.e., buried residues are too variable or exposed residues too conserved). When comparing the amino acid frequencies in the designed proteins with those in natural proteins we find that in the designed proteins hydrophobic residues are underrepresented in the core. From these results we conclude that intermediate backbone flexibility during design results in more accurate protein design and that either scoring functions or backbone sampling methods require further improvement to accurately replicate structural constraints on site variability.

Introduction

Computational protein design has made tremendous progress in recent years. For example, computational design has been used successfully to engineer proteins that bind to an influenza virus (Fleishman et al., 2011), to create enzymes (Röthlisberger et al., 2008), and to develop novel protein folds not seen in nature (Kuhlman et al., 2003). All these examples have in common that many different computational predictions were generated, and among the best were a few that worked experimentally. Thus, while computational design can produce specific sequences that fold correctly and are functional, it is much less clear how similar designed proteins are on average to natural proteins of a comparable fold.

There are several patterns of sequence variation that are consistently seen in natural proteins. For example, amino acid frequencies follow characteristic distributions, and these distributions differ for surfaces and cores of proteins (Overington et al., 1992; Porto et al., 2004; Bastolla et al., 2005; Moelbert, Emberly & Tang, 2004). In particular, hydrophobic residues tend to be more frequent in the core and polar residues tend to be more frequent on the surface. Further, sites in the core of a protein tend to be more conserved and to evolve slower than surface sites (Mirny & Shakhnovich, 1999; Goldman, Thorne & Jones, 1998; Bustamante, Townsend & Hartl, 2000; Conant & Stadler, 2009; Franzosa & Xia, 2009; Ramsey et al., 2011; Scherrer, Meyer & Wilke, 2012; Meyer & Wilke, 2012). Presumably, sites in the core tend to be conserved because mutations at these sites are more likely to destabilize the protein fold, due to steric clashes (Chothia & Finkelstein, 1990).

However, protein properties also vary systematically with factors related to the cellular environment in which proteins are expressed. For example, more highly expressed proteins tend to be more soluble and have less-sticky surfaces (Tartaglia et al., 2007; Levy, De & Teichmann, 2012). Current protein design algorithms optimize primarily for fold stability (Kuhlman & Baker, 2000). Therefore, we would not expect them to reproduce any patterns caused by the cellular expression environment. By contrast, any patterns that are driven primarily by the requirement for sufficient fold stability, such as avoidance of steric clashes in the core, should be reproduced in computationally designed proteins.

Here, we carried out a systematic comparison between alignments of natural sequences and the corresponding alignments of designed sequences, for several different design conditions. We considered two distinct data sets, one of whole protein structures and one of individual protein domains. We analyzed which design conditions produced sequence alignments that were most similar to natural sequence alignments. We also analyzed by which parameters designed proteins differed the most from natural sequences. Overall, we found that proteins designed with a flexible backbone and using an intermediate amount of backbone flexibility were the most similar to natural proteins. However, substantial differences between designed and natural proteins remained even under the most advantageous design conditions. In particular, designed proteins tended to have too many polar and too few hydrophobic residues in the core, and they also tended to have cores that were too variable and/or surfaces that were too conserved. These trends were exacerbated for longer proteins.

Methods

Data sets

We analyzed two data sets, one of whole yeast proteins and one of protein domains. The yeast-proteins data set was taken from Ramsey et al. (2011) and comprised 38 protein structures homologous to an open reading frame in Saccharomyces cerevisiae. For each of those structures, we had at least 50 homologous natural sequences, also taken from Ramsey et al. (2011). The protein-domain data set was taken from Ollikainen & Kortemme (2013) and comprised 40 protein domains. Only domains with at least one crystal structure in the Protein Database (PDB) and at least 500 sequences in the Pfam Database were selected for this data set. Domains were selected in order to represent several different types of protein folds and domains were also restricted to a length less than or equal to 150 amino acids. For each of these protein domains, we obtained alignments of homologous natural sequences from the Pfam database (Punta et al., 2012), as described (Ollikainen & Kortemme, 2013).

Protein design

For each structure in both data sets, we computationally designed 500 variants each, using multiple design methods. All design methods we used are implemented in the protein-design software Rosetta (Leaver-Fay et al., 2011). First, we used standard fixed-backbone design (Kuhlman et al., 2003). In this method, the protein backbone remains fixed and only amino-acid side chains are allowed to move. Second, we used the flexible-backbone method Backrub (Smith & Kortemme, 2008), which first generates an ensemble of alternative backbones and then designs side chains onto these backbones (Friedland et al., 2009; Smith & Kortemme, 2010). The Backrub method takes as input a temperature parameter that determines the extent of backbone movements that occur during design. Here, we used temperatures spanning from 0.03 to 2.4 corresponding to increasing backbone movements. For the protein-domain data set, we also used one additional design method, called “Soft”. This method keeps the backbone fixed but the energy function used during sequence design dampens the weight of the repulsive Lennard-Jones (LJ) potential term (Ollikainen & Kortemme, 2013). Protein designs for the protein-domain data set have been previously published (Ollikainen & Kortemme, 2013), while the designs for the yeast-proteins data set were newly generated for the present study.

All designs for the yeast-proteins data set were generated with Rosetta Revision 39284. For fixed-backbone design, we used the following command:

./fixbb.linuxgccrelease -database rosetta_database \ -s input.pdb -resfile ALLAA.res -ex1 -ex2 \ -extrachi_cutoff 0 -nstruct 1 -linmem_ig 10

Flexible-backbone design was performed by generating a conformational ensemble of 500 structures and then using fixed-backbone design to predict a low energy sequence for each structure in the ensemble. To generate structures for the conformational ensemble, we used the following command:

./backrub.linuxgccrelease -database rosetta_database \ -s input.pdb -resfile NATAA.res -ex1 -ex2 \ -extrachi_cutoff 0 -backrub:mc_kt <T> \ -backrub:ntrials 10000 -nstruct 1 -backrub:initial_pack

where <T> has to be replaced by the desired design temperature.

The design details for the protein-domain data set can be found in Ollikainen & Kortemme (2013).

Data analysis

We quantified the variability of sites in amino-acid alignments using site entropy Hi, defined as Hi=∑jpijlnpij. Here, pij is frequency of amino acid j in alignment column i, and the sum runs over all amino acids. We compared amino-acid distributions of designed sequences to those of natural sequences using the Kullback-Leibler (KL) divergence. The KL divergence DiKL is defined as DiKL=∑jpijln(pij/qij), where qij is the frequency of amino acid j in column i of the reference alignment, and pij is the corresponding frequency in the alignment that is being compared to the reference alignment. The sum runs over all amino acids. When calculating frequencies used for the KL divergence we corrected for the presence of frequencies of zero by adding 1/20 to each amino acid count before calculating the frequencies. The KL divergence is inherently an asymmetric distance measure, comparing a probability distribution of interest to a reference distribution. Unless noted otherwise, we always used natural sequence alignments to calculate the reference frequencies qij and designed sequence alignments to calculate the frequencies pij. Throughout this work, we calculated DiKL separately at each site i in a protein, and then averaged the DiKL values for all sites in a protein to obtain a mean KL divergence for that protein.

To compare the shapes of amino-acid distributions while disregarding specific amino-acid identities, we performed a second type of KL calculation where we ordered amino-acids by their relative frequencies. Thus, instead of the frequencies pij and qij we used pirj and qisj, where rj is the rank of the frequency of amino acid j in column i of the alignment being compared to the reference, and sj is the rank of the frequency of amino acid j in column i of the reference alignment. This way of calculating the KL divergence compares the frequencies of amino acids at equal frequency rank, regardless of which specific amino acids are the most frequent, second-most frequent, and so on in each alignment. As an example, assume that at a given site there are only three different amino acids in the natural alignment, I, L, and V, at frequencies 0.5, 0.35, and 0.15, respectively. At the same site in the designed sequences, there are amino acids A, V, and I, also at frequencies 0.5, 0.35, and 0.15, respectively. In our calculation of KL divergence comparing amino acids at equal frequency rank, we would then compare the frequency of I in the natural alignment with the frequency of A in the designed alignment (the two most frequent amino acids in the two respective alignments) and similarly the frequency of L with the frequency of V and the frequency of V with the frequency of I, respectively. In this example, since the two sets of three frequencies are exactly the same if we disregard amino-acid type, we would obtain a KL divergence of zero.

We calculated Relative Solvent Accessibility (RSA) of residues by first calculating the absolute Solvent Accessibility (ASA) for each residue, using the software DSSP (Kabsch & Sander, 1983). For each protein, we extracted the chain of interest from the PDB structure and ran DSSP only on that chain. We calculated RSA by dividing the ASA value for each residue by the maximum possible ASA value, as given by Tien et al. (2013). Throughout this work, we only calculated RSA on the native PDB structure. We did not perform any RSA calculations on designed structures.

All our data and analysis scripts are available online at: https://github.com/clauswilke/protein_design_and_site_variability.

Results

We wanted to assess the extent to which the sequence space of computationally designed proteins overlaps with the sequence space occupied by homologous natural proteins. Our general approach was to compare alignments of designed protein sequences to alignments of homologous natural sequences, for approximately 80 distinct protein structures. For each structure, we considered several different design methods (see Methods for details), and we designed 500 sequences for each structure and method. The protein structures we considered were subdivided into two distinct data sets, a data set of 38 yeast protein structures previously analyzed by Ramsey et al. (2011) and a data set of 40 protein domains previously analyzed by Ollikainen & Kortemme (2013). Throughout this study, we analyzed these two data sets separately, because they corresponded to structures of substantially different sizes. The mean number of amino acids per structure was 215.4 in the yeast-proteins data set and 86.1 in the protein-domains data set. Also, the overall sequence variability of the protein-domain data set was greater than the variability of the yeast-proteins data set.

Overall site variability

We first compared overall amino-acid variability in designed and natural proteins. We assessed amino-acid variability at individual sites by calculating the entropy Hi at each site i in alignments of either designed or natural proteins. We then calculated the mean entropy over all sites in each alignment and used that quantity as a measure of the overall amino-acid variability in the alignment.

We found that protein design using a fixed backbone generally yielded insufficient site variability compared to natural sequences (Fig. 1). This result was magnified in the smaller protein domains. In fact, for the protein domains, the most variable proteins under fixed-backbone design showed only about as much variability as the least variable natural proteins. Overall, there was a significant shift towards higher variability in natural proteins relative to proteins designed with fixed backbone (paired t test, P = 1.4 × 10−10 for the yeast-proteins data set and P < 10−15 for the protein-domain data set). When switching from fixed-backbone design to variable-backbone design, we found that overall site variability increased. Further, site variability increased monotonously with the degree of backbone flexibility allowed during design, as measured by the design temperature (Fig. 1). At the highest temperatures, site variability in designed proteins consistently exceeded that of natural proteins.

Figure 1 Mean site entropy for designed and natural proteins.

Each boxplot represents the distribution of mean site entropies within the respective dataset ((A) yeast proteins; (B) protein domains). “FB” refers to fixed-backbone design. Temperature values refer to the design temperature used during the Backrub design method. “NS” refers to natural sequences. “Soft” refers to the Soft design method. We find generally that increased backbone flexibility allows for more site variability. Intermediate temperatures produce site variabilities most similar to those seen in natural sequences. Overall, natural sequences in the protein-domains data set are more variable than are those in the yeast-proteins data set.

Proteins designed at intermediate temperatures had site variabilities that most closely resembled that of natural proteins. For the yeast-proteins data set, the temperature that provided the closest match was T = 0.03, even though the variability of sequences designed at that temperature still exceeded the variability in natural sequences (paired t test, P = 0.0006). For the protein-domains data set, the temperature that provided the closest match was T = 0.9, for which variability was statistically indistinguishable from that found in natural sequence alignments (paired t test, P = 0.353). However, for both data sets, natural sequences generally showed a larger spread in variabilities than did the designed sequences at the closest-matching temperatures (Brown-Forsythe test for equal variances, P = 0.0003 for the yeast-proteins data set at T = 0.03 and P = 7.3 × 10−6 for the protein-domain data set at T = 0.9).

Amino-acid distributions

We next compared amino-acid distributions between designed and natural sequences. First we looked at overall amino acid frequencies. We found that by-and-large, amino acid frequencies in designed proteins mirrored those in natural proteins (Fig. 2 and Figs. S1–S5). The biggest differences arose in Pro, His, Trp, Phe, and Ala. (We ignore Cys here because Cys is never used in the design algorithm and thus always at frequency 0.) Overall, we observed that hydrophobic residues tended to be under-represented in designed proteins whereas hydrophilic residues tended to be over-represented. This trend was stronger in the protein core than on the surface. We also observed that the longer proteins in the yeast-proteins data set showed larger deviations between designed and natural sequences than the shorter proteins in the protein-domains data set. Finally, when comparing different design methods and design temperatures, we found that differences in amino-acid distributions were relatively minor, see Fig. 2 and Figs. S1–S5.

Figure 2 Amino-acid frequencies in designed and natural proteins.

Frequencies were calculated over all sites in all proteins belonging to the yeast-proteins data set. For designed proteins, only flexible-backbone designs with design temperature 0.6 were considered. (A) Overall frequencies. (B) frequencies at exposed sites (defined as sites with RSA > 0.05). (C) frequencies at buried sites (defined as sites with RSA ≤ 0.05).

Even if overall amino-acid distributions are approximately correct, the amino-acid distributions at individual sites can be poorly predicted (Ramsey et al., 2011). Therefore, we next compared, separately at each site, the similarity between amino-acid distributions in natural proteins and those in designed proteins. To carry out this comparison, we employed the Kullback-Leibler (KL) divergence (Wasserman, 2004), which measures how similar one probability distribution is to a reference distribution. A KL divergence of zero implies that the distributions are identical. The higher the KL divergence, the more dissimilar the focal distribution is to the reference distribution. (Note that KL divergence is not symmetric: if we swap the focal and the reference distribution, we will generally obtain a different KL divergence value.) We calculated the KL divergence at each site in each protein, and then averaged over sites within a protein to obtain a mean similarity score for each protein. As a control, we also randomly split the alignment of natural sequences for each protein structure into two halves and calculated the mean KL divergence of natural sequences against themselves.

First, in all comparisons, we found that the KL divergence of designed relative to natural sequences was much bigger than the KL divergence of natural sequences relative to themselves (Fig. 3 and Fig. S6). This finding indicates a substantial discrepancy between designed and natural sequences at individual sites. Second, we found that the mean KL divergence decreased with increasing design temperature (Fig. 3A and Fig. S6A). Thus, according to the KL divergence measure, structures designed with the most flexible backbones had the most similar amino-acid distributions to those found in natural sequences.

However, the result that sequences designed at the highest temperatures are the most similar to natural sequences may be an artifact of the KL divergence measure. As design temperature increases, amino-acid variability increases, and amino-acid distributions become more uniform. A more uniform distribution is generally going to display more overlap with any given distribution than a more localized distribution, if the localized distribution is not correct. Thus, the decrease in KL divergence with increasing temperature may simply reflect the broadening of the distribution, not an actual improvement in reproducing natural amino-acid distributions. To assess whether amino-acid distributions in designed sequences were simply broadening with increasing temperature, or whether they were actually converging on the natural distributions, we carried out a second set of comparisons. We rank-ordered amino acids by frequency at each site in each protein, and then calculated the KL divergence of the rank-ordered distributions.

This comparison considers only the shape of the distribution and does not assess whether the correct amino acids are present at individual sites. This second comparison generally found much lower KL divergence levels, even though still not as low as what was found for the control comparison of natural sequences with themselves (Fig. 3B and Fig. S6B). More importantly, now KL divergence reached a minimum around a temperature of 0.3 (yeast proteins, Fig. 3B) to 1.2 (protein domains, Fig. S6B) and rose again beyond that value. This finding indicates that higher design temperatures do not unequivocally produce more natural amino-acid distributions. Instead, there is an intermediate temperature, approximately coinciding with the temperature at which overall sequence variability matches best, at which amino acid distributions also are most similar.

Figure 3 Mean Kullback-Leibler (KL) divergence for designed and natural proteins, shown for the protein-domain data set.

A higher KL divergence indicates that the amino-acid distributions at sites in designed proteins are less similar to the corresponding distributions in the natural proteins. “FB” refers to fixed backbone design and “NS” refers to the control case where natural sequences are compared to themselves. (A) KL divergence calculated from the relative frequencies of the 20 amino acids. (B) KL divergence calculated from rank-ordered frequency distributions. The most common amino acid in the reference distribution is compared to the most common amino acid in the focal distribution, the same is done for the second-most common amino acid, and so on, irrespective of the type of amino acids.

Site variability and solvent accessibility

The previous analyses demonstrated that while designed proteins overall look similar to natural proteins, there are also important differences. We next wanted to identify whether these differences were present uniformly throughout the structure or could be located to specific structural regions. In our analysis of amino-acid distributions, we had already seen that amino-acid distributions seemed to deviate more at buried sites than at exposed sites (Fig. 2 and Fig. S4).

Figure 4 Distributions of correlation coefficients between site entropy and RSA, for the protein-domain data set.

“FB” indicates fixed-backbone design and “NS” indicates natural sequences. (A) Distributions represented as boxplots. (B) Correlation coefficients for individual proteins. Lines connect identical structures in the different design conditions. The color shading represents the strength of the correlation for the natural sequence alignment. In general, natural proteins display a stronger correlation between site entropy and RSA than designed proteins.

We first plotted site variability against relative solvent accessibility (RSA, a dimensionless number from 0 to 1 measuring the relative solvent exposure of individual residues) for individual proteins. See Fig. S7 for one example. We generally found that site variability displayed a substantial spread even for sites of very similar RSA. At the same time, there was an overall trend for sites with higher RSA to be more variable than sites with lower RSA. This trend was generally stronger in flexible backbone designs than in fixed backbone designs (Fig. S7). To analyze the relationship between site variability and RSA more systematically, we calculated the correlation between these two quantities for all proteins (Fig. 4 and Fig. S8). On average, natural sequence alignments showed a higher correlation than alignments of designed sequences, regardless of design method.

Intermediate design temperatures showed the highest correlations, but correlations were nevertheless significantly lower in designed proteins than in natural proteins (paired t test, P = 2.96 × 10−10 [T = 0.3, yeast proteins] and P = 1.75 × 10−5 [T = 0.3, protein domains]). We also investigated whether the designed proteins with the highest correlations corresponded to the natural proteins with the highest correlations, and found this generally to be the case (Fig. 4B and Fig. S8B).

Our finding that correlations between site entropy and RSA are lower in designed proteins than in natural proteins indicates that, in designed proteins, site variability is too uniform across different solvent exposure states. In short, designed proteins are either too variable in the core or too conserved on the surface. To obtain a clearer picture of how exactly designed proteins differed from natural proteins, we once more considered the distributions of mean site entropies, but now calculated separately for buried sites (RSA ≤ 0.05), for partially buried sites (0.05 < RSA ≤ 0.25), and for exposed sites (RSA > 0.25). Figure 5 shows the medians of these distributions. For designed proteins, the mean site variabilities of exposed and of partially buried sites are close in magnitude while the mean site variabilities of buried sites are generally consistently lower. By contrast, in natural sequences exposed sites show much more variability than partially buried sites.

Figure 5 Median of the distribution of mean sequence entropies for designed and natural sequences, calculated separately for buried (black), partially buried (blue), and exposed (red) residues.

We defined buried sites as those with RSA ≤ 0.05, partially buried as those with 0.05 < RSA ≤ 0.25, and exposed as those with RSA > 0.25. Dashed lines indicate the corresponding median for natural sequence alignments. Note that for buried (black) and partially buried (blue) residues, the temperatures at which natural site variability and design variability match are comparable. By contrast, for exposed residues, a higher design temperature is required for the design variability to match the natural site variability. (A) yeast proteins; (B) protein domains.

If buried sites are too variable or exposed sites too conserved in designed proteins, we reasoned that hybrid designs, in which buried sites were taken from sequences designed at a lower temperature and exposed sites from sequences designed at a higher temperature, should display correlations more similar to those seen in natural proteins.

According to Fig. 5, for the yeast proteins buried and partially buried sites in designed proteins had site variability most similar to that of natural sequences in proteins designed with a fixed backbone or in proteins with a design temperature of T = 0.03. In the protein-domains data set, that temperature was T = 0.3 to T = 0.6. By contrast, for exposed sites the site variability in designed proteins was most similar to that of natural sequences at a design temperature of T = 0.1 (yeast proteins) and T = 1.2 (protein domains). We thus built our hybrid designs by combining sites from these temperatures. We found that the distribution of the site-entropy–RSA correlations in hybrid designs was comparable to that in natural sequences (Fig. 6). However, predictions for specific proteins lacked accuracy (Fig. S9).

Figure 6 Distribution of correlation coefficients between RSA and site entropy for hybrid designs and for natural proteins.

“FB” indicates fixed-backbone design and “NS” indicates natural sequences. For the hybrid designs, buried and partially buried sites were taken from sequences designed at one temperature, and exposed sites were taken from sequences designed at a different temperature. For the hybrid designs, the correlation coefficients were similar to those of natural sequences (paired t test, P = 0.517 [T = FB, 0.1] and P = 6.78 × 10−8 [T = 0.03, 0.1], yeast proteins, and P = 5.19 × 10−5 [T = 0.3, 1.8] and P = 0.118 [T = 0.6, 1.8], protein domains). (A) yeast proteins; (B) protein domains.

Discussion

We have compared site variability and amino-acid distributions in designed and natural proteins, for two distinct data sets. One data set consisted of 38 yeast proteins and the other consisted of 40 protein domains. Structures in the yeast-proteins data set were, on average, much larger than structures in the protein-domain data set, while natural sequences in the protein-domain data set were more variable than those in the yeast-proteins data set. We have found that proteins designed with a flexible backbone, using an intermediate design temperature, were generally the most similar to natural proteins. Overall amino-acid frequencies in designed proteins were similar, though not identical, to those in natural proteins. However, amino-acid frequencies at individual sites showed substantial deviations. Finally, we have found that site variabilities in designed proteins are too uniform across different solvent exposure states of residues. Designed proteins have either cores that are too variable or surfaces that are too conserved.

In previous studies, native sequence recovery has been used to assess design accuracy. (Gainza, Roberts & Donald, 2012; Kuhlman & Baker, 2000). Native sequence recovery is defined as the mean percent of native amino acid identities that are observed in the designed proteins. Despite its widespread use, native sequence recovery may not always be a sufficient indicator of design accuracy, especially when examining different sequences that are compatible with one specific structure. A major goal of design is to find sequences that fold into a specific structure. For this goal, one typically models a series of structures that are similar to the native structure and then identifies low energy sequences for each of these modeled structures. Even if all designed sequences fold into the desired structure, they may not necessarily have a high sequence similarity with the sequence of the native structure. For this reason, we believe that it is important to assess design accuracy by multiple different methods, and also against an ensemble of native sequences or structures.

A previous study, the source of the protein-domains data set we analyzed here, has similarly compared designed proteins against ensembles of natural sequences (Ollikainen & Kortemme, 2013). That study and our present study complement each other. Ollikainen & Kortemme (2013) were primarily interested in amino-acid covariation, and they also considered sequence entropy and profile similarity (Yona & Levitt, 2002). Here, we were primarily interested in the effects of solvent occlusion on site variability and amino-acid choice, and we also considered two distinct sets of natural reference structures (protein domains and whole proteins). In both studies, an intermediate amount of backbone flexibility was found to be optimal for recapitulating characteristics of natural protein sequences. Both studies also identify similar inaccuracies in the designed protein sequences. Ollikainen & Kortemme (2013) observed that covarying pairs in designed protein cores were more likely to be hydrogen bonding pairs that in natural cores, and here we found that polar residues are over-represented in the designed protein cores compared to natural cores.

Our analysis compared two distinct datasets. The first was comprised of 40 protein domains, chosen to be less than 150 amino acids in length and with a mean length of 86.1 amino acids. The second was comprised of 38 whole yeast proteins, with a mean length of 215.4 amino acids. For each structure in each data set, we had an associated alignment of natural sequences to assess natural variability for that structure. (Note that sequences homologous to the yeast proteins were not constrained to be fungal sequences.) Sequence in the protein-domain data set were more variable than sequences in the yeast-protein data set. We found that optimal design temperatures were lower for the yeast-protein data set than for the protein-domains data set. This finding is consistent with both increased mean length and reduced mean variability in the yeast-protein data set relative to the protein-domains data set. In particular, large cores in the larger proteins may lead to larger conserved regions whose site variability patterns are better recaptured at lower design temperatures.

We found that the characteristics of designed protein sequences are generally similar but by no means identical to natural sequences. To some extent, this discrepancy is to be expected. Designed protein sequences are optimized entirely for thermodynamic stability as estimated by the design energy function. Natural proteins experience a variety of selective pressures, stability being only one of them. For example, natural proteins experience selection pressures for native protein–protein interactions, against non-specific protein–protein interactions, and against misfolding and aggregation (Fraser et al., 2002; Zarrinpar, Park & Lim, 2003; Drummond & Wilke, 2008; Levy, De & Teichmann, 2012). If they are enzymes, natural proteins also require the appropriate mutations that enable enzymatic activity, even if those mutations are thermodynamically destabilizing (Bloom et al., 2006; Elcock, 2001). While selection for enzymatic activity will likely affect only a few sites in a protein, the other selective forces (misfolding, aggregation, native and non-specific interactions) have the potential to exert much broader selection pressures across many sites in a protein. As long as design algorithms do not take these selection pressures into account, we cannot expect design algorithms to reproduce natural sequence variation exactly.

To identify at which sites discrepancies between natural and designed proteins arose, we explicitly examined the relationship between structure and sequence variability. In particular, we analyzed the correlation between RSA and site entropy, which reflects the well-known observation that proteins are more variable on the surface than in the core. We found that the difference between surface and core variability was much more pronounced in natural proteins than in designed proteins. Designed proteins either have cores that are too variable or surfaces that are too conserved. We created hybrid designs, taking core sites from one set of designed proteins and surface site from another set, designed with more backbone movement, and tested whether these hybrid designs showed the appropriate differential in variability between core and surface sites. We found that they did so as a population (Fig. 6) but not individually (Fig. S9). This observation indicates that there is some aspect of protein fold stability that differentially affects surface and core residues and that is not yet properly incorporated into current design algorithms. Simply raising the design temperature on the surface but not in the core is not sufficient to capture this effect. Note that we do not expect our hybrid design approach to yield realistic, stable protein sequences. It is merely meant as an illustration of the extent to which surface sites would have to be more variable relative to core sites to yield entropy-RSA correlations comparable to those found in natural sequences.

For both data sets, the designed proteins had fewer hydrophobic residues and more polar residues than expected from natural sequence alignments. This trend was particularly apparent in the protein core, and it was more extreme for larger proteins. These discrepancies suggest a need for further improvement of the design algorithm, most likely the energy function. Rosetta uses a scoring function that predicts the energy of a given sequence folded into a particular target structure (Kuhlman et al., 2003). As a component of this scoring function, Rosetta uses the Lazaridis-Karplus implicit solvation model to estimate the energy of desolvation of each residue (Lazaridis & Karplus, 1999). The over-representation of polar residues in protein cores that we observed suggests that this solvation model is either insufficiently penalizing for the burial of polar groups or insufficiently rewarding the burial of hydrophobic residues. Improvements to the solvation model used in design may result in more stable designed proteins with amino acid distributions more similar to those of natural proteins, especially in protein cores.

While protein cores are more variable in designed proteins compared to natural proteins, the surfaces of designed proteins are too conserved. This discrepancy is somewhat expected. We would only expect close agreement between designed and natural proteins if the sequences are under the same constraints (and provided the energy function could accurately capture these). Computational design optimizes sequences primarily for protein stability, which, in natural proteins, is more likely to be a dominant constraint in protein cores than on surfaces. Surfaces of natural proteins may also be under other important pressures, such as to make desired and avoid unwanted interactions and to keep proteins soluble. All of these pressures could act to diversify protein surfaces away from sequence choices that would maximize stability. In addition, there are of course also inaccuracies in the design energy function, including difficulties in accurately modeling electrostatics and solvation at surfaces, and contributions of conformational entropy of surface side chains that are not taken into account in most design energy functions.

In our analysis of approximately 80 protein structures total, we found that proteins designed with an intermediate amount of backbone flexibility exhibited site-variability patterns most closely resembling that of natural proteins. However, the optimal range of backbone flexibility was different in the two data sets. Further, even when the overall site variability matched that of natural sequences, the specific amino-acid distributions at individual sites did not match that well, as quantified by the relatively large KL divergence values between natural and designed alignments. Similarly, intermediate design temperatures showed the highest correlation between RSA and site variability (as measured by entropy). However, even at the optimal design temperature (T ∼ 0.3 for both data sets), the designed proteins exhibited systematically lower correlations than did the natural proteins. Consequently, using current state-of-the-art design algorithms, designed proteins have either surfaces that are too conserved or cores that too variable. We suspect that changes in the design energy function, in particular more accurate estimation of the balance between electrostatics and desolation, will be needed to address this issue. We also see a need for improved flexible-backbone design algorithms that can model larger backbone movements on the surface without disturbing the core backbone as much. As alternative and improved algorithms design algorithms become available, they should be subjected to similar tests as we have done here, to assess to what extent different algorithms reproduce natural amino-acid frequency and site-variability differences in core versus surface.

Supplemental Information

Figure S1 Amino-acid frequencies in designed and natural proteins

Frequencies were calculated over all sites in all proteins belonging to the yeast-proteins data set. For designed proteins, only fixed-backbone designs were considered. (A) overall frequencies. (B) frequencies at exposed sites (defined as sites with RSA > 0.05). (C) frequencies at buried sites (defined as sites with RSA ≤ 0.05).

Click here for additional data file.

Figure S2 Amino-acid frequencies in designed and natural proteins

Frequencies were calculated over all sites in all proteins belonging to the yeast-proteins data set. For designed proteins, only flexible-backbone designs with design temperature 1.2 were considered. (A) overall frequencies. (B) frequencies at exposed sites (defined as sites with RSA > 0.05). (C) frequencies at buried sites (defined as sites with RSA ≤ 0.05).

Click here for additional data file.

Figure S3 Amino-acid frequencies in designed and natural proteins

Frequencies were calculated over all sites in all proteins belonging to the protein-domains data set. For designed proteins, only fixed-backbone designs were considered. (A) overall frequencies. (B) frequencies at exposed sites (defined as sites with RSA > 0.05). (C) frequencies at buried sites (defined as sites with RSA ≤ 0.05).

Click here for additional data file.

Figure S4 Amino-acid frequencies in designed and natural proteins

Frequencies were calculated over all sites in all proteins belonging to the protein-domains data set. For designed proteins, only flexible-backbone designs with design temperature 0.6 were considered. (A) overall frequencies. (B) frequencies at exposed sites (defined as sites with RSA > 0.05). (C) frequencies at buried sites (defined as sites with RSA ≤ 0.05).

Click here for additional data file.

Figure S5 Amino-acid frequencies in designed and natural proteins

Frequencies were calculated over all sites in all proteins belonging to the protein-domains data set. For designed proteins, only flexible-backbone designs with design temperature 1.2 were considered. (A) overall frequencies. (B) frequencies at exposed sites (defined as sites with RSA > 0.05). (C) frequencies at buried sites (defined as sites with RSA ≤ 0.05).

Click here for additional data file.

Figure S6 Mean Kullback-Leibler (KL) divergence for designed and natural proteins, shown for the yeast-proteins data set

A higher KL divergence indicates that the amino-acid distributions at sites in designed proteins are less similar to the corresponding distributions in the natural proteins. “FB” refers to fixed backbone design, and “NS” refers to the control case where natural sequences are compared to themselves. (A) KL divergence calculated from the relative frequencies of the 20 amino acids. (B) KL divergence calculated from rank-ordered frequency distributions. The most common amino acid in the reference distribution is compared to the most common amino acid in the focal distribution, the same is done for the second-most common amino acid, and so on, irrespective of the type of amino acids.

Click here for additional data file.

Figure S7 Site entropy versus Relative Solvent Accessibility (RSA) for designed and natural sequence alignments of the protein S-formylglutathione hydrolase (PDB: 1PV1, chain A)

Natural sequences exhibit a clear trend of higher site variability at higher RSA values. The flexible backbone designs exhibit a similar trend but the fixed backbone designs do not.

Click here for additional data file.

Figure S8 Distributions of correlation coefficients between site entropy and RSA, for the yeast-proteins data set

“FB” indicates fixed-backbone design, “Soft” indicates soft backbone design, and “NS” indicates natural sequences. (A) Distributions represented as boxplots. (B) Correlation coefficients for individual proteins. Lines connect identical structures in the different design conditions. The color shading represents the strength of the correlation for the natural sequence alignment. In general, natural proteins display a stronger correlation between site entropy and RSA than designed proteins.

Click here for additional data file.

Figure S9 Correlation coefficients between RSA and site entropy for hybrid designs and natural proteins

For the hybrid designs, buried and partially buried sites were taken from proteins designed with a fixed backbone (yeast proteins) or a temperature of T = 0.6 (protein domains). Exposed residues were taken from proteins designed with a temperature of T = 0.1 (yeast proteins) or T = 1.8 (protein domains). The solid line indicates y = x. Note that while the range of correlation values in hybrid designs generally matches the range of values in natural alignments, predictions for specific proteins are not that accurate.

Click here for additional data file.

The authors acknowledge the Texas Advanced Computing Center (TACC) at The University of Texas at Austin for providing HPC resources that have contributed to the research results reported within this paper.

Additional Information and Declarations

Competing Interests

Author Contributions

Data Deposition

The authors declare they have no competing interests.

Eleisha L. Jackson conceived and designed the experiments, performed the experiments, analyzed the data, wrote the paper.

Noah Ollikainen and Arthur W. Covert III conceived and designed the experiments, performed the experiments, contributed reagents/materials/analysis tools, wrote the paper.

Tanja Kortemme and Claus O. Wilke conceived and designed the experiments, wrote the paper.

The following information was supplied regarding the deposition of related data:

GitHub: https://github.com/clauswilke/protein_design_and_site_variability.

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
