# Peer review of "Amino-acid site variability among natural and designed proteins"

_PeerJ, doi:10.7717/peerj.211_

## Round 0.1 · original submission · Minor Revisions

Please address critical points raised by both reviewers.

Reviewer 1 ·

Basic reporting

This study describes an analysis of how well different design protocols (that incorporate an increasing degree of backbone flexibility) reproduce native sequence profiles (from PFAM). Two sets are analyzed: one of yeast proteins, the other of domains.

The subject is well introduced, and a basic background is given.

Overall basic reporting adheres to the guidelines of PeerJ

Experimental design

In a large scale redesign study, different proteins are redesigned under different conditions (e.g. fixed backbone, soft design, and backrub-based design - using increasing temperatures to obtain increased variability of the backbone conformations).

I have one minor comment regarding the evaluation of differences between designed and natural sequences using a rank-order instead of actual frequencies. The authors correctly observe that with increased numbers of sequences, the distributions of amino acid frequencies at a given position tend to get flatter, and thus differences cold arise. They therefore apply a different measure that should get rid of these biases by comparing ranks rather than values. The explanation of the authors as to what exactly is compared should be clarified more. In principle, this looks to me like using a non-parametric rank test, but the details indicate that this is not the case.

In particular, the following sentence is not clear:

"This way of calculating the KL divergence compares the frequencies of amino acids at equal frequency rank, regardless of which specific amino acids are the most frequent, second-most frequent, and so on in each alignment."

Also, no details of the parameters of the rosetta design runs are included. Which version of rosetta was used, and what were the specific flags? E.g., did the authors use additional rotamers?

Validity of the findings

The main findings show that incorporation of backbone flexibility increases sequence variability (which was too low in fixed backbone design compared to natural sequences), but if too much flexibility is added, the variability is too much increased. A suggestion of some hybrid approach is presented that uses different degrees of backbone flexibility to design exposed or buried residues.

the way the two are combined might be problematic as redesign of one position is influenced by the neighboring positions.

Reviewer 2 ·

Basic reporting

Although computational protein design generate sequences that maximize the structural similarity w.r.t a given native fold, functionally very few of those designed sequences mirror the native fold. So, a detailed study of the extent of similarity/dissimilarity between the designed and the natural sequences can be informative for improvement of existing protein design algorithms. The authors have framed this question of interest in a lucid way and appropriate background have been provided.

Experimental design

Both data presentation and data analysis have been extremely thorough and sound and connects/address the stated question.The authors consider two distinct data sets, one of yeast proteins and one of yeast protein domains. Through a systematic comparison between the alignments of designed and natural sequences, the authors have teased apart how sequence patterns/features in the former differ from the latter under different design conditions, such as backbone flexibility.

Validity of the findings

All the data have been interpreted and particularly, the authors have done a great job in providing detailed physical explanations of all the data/result, be it expected or unexpected. The discussion provides lots of crucial insights into things that are needed to be considered for improvement of protein design algorithm, e,g. inclusion of selection pressure, modification of solvation term etc.

Additional comments

The authors should use some other protein design tools that use different force fields and algorithms than RosettaDesign. This will show how much the observed difference in sequence pattern is robust or dependent on the algorithm and force-field.

---

## Round 0.2 · accepted · Accept

I beleive that you adequately addressed the comments of both reviewers and therefore the manuscript can be accepted in its present form.